# A Simple Recurrent Unit with Reduced Tensor Product Representations

## Abstract

Widely used recurrent units, including Long-short Term Memory (LSTM) and the Gated Recurrent Unit (GRU), perform well on natural language tasks, but their ability to learn structured representations is still questionable. Exploiting reduced Tensor Product Representations (TPRs) — distributed representations of symbolic structure in which vector-embedded symbols are bound to vector-embedded structural positions — we propose the TPRU, a simple recurrent unit that, at each time step, explicitly executes structural-role binding and unbinding operations to incorporate structural information into learning. A gradient analysis of our proposed TPRU is conducted to support our model design, and its performance on multiple datasets shows the effectiveness of our design choices. Furthermore, observations on a linguistically grounded study demonstrate the interpretability of our TPRU.

## 1 Introduction

Recurrent units are widely used in sequence modelling tasks, including text, speech, DNA/RNA sequencing data, etc. Due to their ability to produce distributed vector representations for sequences with various lengths, and their property of being a universal function approximators (Schäfer & Zimmermann, 2006), they quickly succeeded in numerous research areas. The training algorithm, backpropagation through time (BPTT, Werbos et al., 1990), has remained the same across the years, with known gradient vanishing and exploding issues. Many theoretically justifiable or empirically useful regularisations (Pascanu et al., 2013; Kanai et al., 2017) have been proposed to ease the learning process and to smooth the gradient flow for long sequences, and informative analysis methods have been conducted to visualise the information encoded in the learnt vector representations (Zeiler & Fergus, 2013; Karpathy et al., 2015). The main difficulty in understanding the learning process of a particular RNN unit comes from the extreme nonlinear activation functions, which cause the gradient flow to be chaotic. Here, we aim to propose a recurrent unit that has fewer nonlinear operations by incorporating Tensor Product Representations (Smolensky, 1990).

Tensor product representation (TPR) is an instantiation of general neural-symbolic computing in which symbol structures are embedded in a vector space via a *filler-role decomposition*: the structure is captured by a set of $N$ **roles** $r_i$ (e.g., left-child-of-root), each of which is bound to a **filler** $f_i$ (a symbol or substructure). A TPR embedding of a symbol structure $\boldsymbol{S}$ derives from vector embeddings of the roles $\{\boldsymbol{r}_i\}$ and their fillers $\{\boldsymbol{f}_i\}$ via the outer or tensor product: $\boldsymbol{S} = \sum_{i=1}^{N} \boldsymbol{r}_i \otimes \boldsymbol{f}_i = \sum_{i=1}^{N} \boldsymbol{r}_i \boldsymbol{f}_i^\top = \boldsymbol{R}\boldsymbol{F}^\top$, where $\boldsymbol{R}$ and $\boldsymbol{F}$ respectively denote matrices having the role or filler vectors of $\boldsymbol{S}$ as columns. Each $\boldsymbol{r}_i \otimes \boldsymbol{f}_i$ is the embedding of a role-filler binding: a constituent; $\otimes$ is the **binding** operation. The **unbinding** operation returns the filler of a particular role in $\boldsymbol{S}$; it is performed by the inner product: $\boldsymbol{f}_i = \boldsymbol{u}_i^\top \boldsymbol{S}$, where $\boldsymbol{u}_i$ is the dual of $\boldsymbol{r}_i$, satisfying $\boldsymbol{u}_i^\top \boldsymbol{r}_j = \delta_{ij} \equiv 1$ if $i = j$ else 0. Letting $\boldsymbol{U}$ be the matrix with columns $\{\boldsymbol{u}_i\}$, we have $\boldsymbol{U}^\top \boldsymbol{R} = \boldsymbol{I}$.

Our goal is to propose a recurrent unit that uses a reduced version of a tensor product representation, a vector rather than an order-2 tensor: this arises by reducing the dimension of the filler embeddings to 1; with role embeddings of dimension $d$, the representation becomes $\boldsymbol{b} = \sum_{i=1}^{N} f_i \boldsymbol{r}_i \in \mathbb{R}^d$. We let this sum range over all $N$ possible roles, letting $f_i = 0$ when role $r_i$ is unfilled. Then at time $t$, the hidden state of our RNN — which we call the **binding complex** — will be $\boldsymbol{b}_t = \sum_{i=1}^{N} (\boldsymbol{f}_t)_i \boldsymbol{r}_i$ where $\boldsymbol{f}_t$ is the vector of filler values $(\boldsymbol{f}_t)_i \in \mathbb{R}$ at time $t$. Now binding is performed by a simple matrix-vector product, and unbinding by a simple inner product:

$$\boldsymbol{b}_t = \boldsymbol{R}\boldsymbol{f}_t \qquad \boldsymbol{f}_t = \boldsymbol{U}^\top \boldsymbol{b}_t \qquad (1)$$

In this paper:

• In the context of related work (Sec. 2) and based on the reduced tensor product representations introduced above, we propose a gated recurrent unit, the **TPRU** (Sec. 3), along with a set of analyses of the gradient flow to justify our decisions in architecture design (Sec. 4). Even though the number of nonlinear operations is reduced, the success in experiments attests to the flexibility of our TPRU in learning and our analysis on derivatives further supports it.

• Our proposed TPRU demonstrates strong performance on the Logical Entailment task (Evans et al., 2018), Natural Language Inference datasets (Williams et al., 2017), and language modelling, while having comparably many parameters as the GRU, and significantly fewer than the widely-used LSTM (Sec. 5).

• The instantiation of reduced TPRs allows our model to learn decomposable representations, and the filler $f_t$ provides an interface for us to inspect the information encoded in $b_t$ (Sec. 6). We conducted a linguistically grounded study, which shows that our TPRU is capable of distinguishing multiple senses of a polysemous word and discovering structured information in sentences.

## 2 RELATED WORK

Recent efforts on understanding RNNs includes these three classes: (i) theoretical analysis through advancements in dynamic systems on the functions learnt by RNN units (Chang et al., 2019); (ii) linguistically grounded analysis by developing sets of probing tasks to illustrate the properties of learnt vector representations of RNNs (Conneau et al., 2018; Chrupała & Alishahi, 2019); and (iii) introducing inductive biases into the architecture of networks and comparing the behaviour of learnt models against those of widely-used counterparts. Our work falls in this last category: learning structured representations by incorporating the inductive biases inherent in TPRs.

Some prior work has incorporated TPRs into RNN representations. Question-answering on SQuAD (Rajpurkar et al., 2016) was addressed (Palangi et al., 2017) with an LSTM with hidden state given by a single TPR binding; the present work deploys complexes containing multiple bindings (albeit reduced ones). TPR-style unbinding was applied in caption generation (Huang et al., 2018), but the representations were deep-learnt and not explicitly designed to be TPRs as in the present work. A contracted version of TPRs, Holographic Reduced Representations, was utilised to decompose input space and output space with filler-role decomposition (Luo et al., 2019). Our work differs from these prior papers in that TPR filler-role binding and unbinding operations are explicitly carried out in our proposed recurrent unit, and the two operations directly determine the update of the hidden states. Closest to the present work is Schlag & Schmidhuber (2018), which performed question-answering by using TPRs to embed simple knowledge graphs as tensors; our model does not deploy graph-based inference, which is less natural for the inference tasks we address than it is for the question-answering task addressed in Schlag & Schmidhuber (2018).

Our proposed TPRU inherits the explicit binding and unbinding operations in TPRs in the sense that a set of role vectors are shared across time steps and sequences to serve as a basis for composing distributed vector representations. We provide a set of analyses from both theoretical and empirical perspectives: gradient analysis of our model helps us make sense of the architecture design, while empirical results from three different categories of tasks show the effectiveness of the proposed model. Additional study on the linguistic and learning aspects provide evidence that our model indeed inherits the merits of TPRs, which enhance interpretability and the capability of learning structured representations.

## 3 PROPOSED RECURRENT UNIT: THE TPRU

As shown in both LSTM (Hochreiter & Schmidhuber, 1997) and GRU (Chung et al., 2014) design, a gating mechanism helps the hidden state at the current time step to directly copy information from the previous time step, and alleviate vanishing and exploding gradient issues. Our proposed recurrent unit deploys a single gate, adopting the input gate of the GRU, omitting the reset gate. We expect that further augmenting our TPRU to 2 gates as in the GRU, or 3 gates as in the LSTM, will lead to yet better performance.

At each time step, the TPRU receives two input vectors: $\boldsymbol{b}_{t-1} \in \mathbb{R}^d$, the binding complex from the TPRU at the previous time-step, and $\boldsymbol{x}_t \in \mathbb{R}^{d'}$ from the external input; it produces $\boldsymbol{b}_t \in \mathbb{R}^d$. An input gate $\boldsymbol{g}_t \in \mathbb{R}^d$ is computed to produce a weighted sum of the information generated at the current time step, $\widetilde{\boldsymbol{b}}_t \in \mathbb{R}^d$, and the previous binding complex $\boldsymbol{b}_{t-1}$,

$$\boldsymbol{b}_t = \boldsymbol{g}_t \circ \widetilde{\boldsymbol{b}}_t + (1 - \boldsymbol{g}_t) \circ \boldsymbol{b}_{t-1}, \text{ where } \boldsymbol{g}_t = \sigma(\boldsymbol{W}_b \boldsymbol{b}_{t-1} + \boldsymbol{W}_x \boldsymbol{x}_t), \tag{2}$$

$\sigma(\cdot)$ is the logistic sigmoid function (applied element-wise), $\circ$ is the Hadamard (element-wise) product, and $\boldsymbol{W}_b \in \mathbb{R}^{d \times d}$ and $\boldsymbol{W}_x \in \mathbb{R}^{d \times d'}$ are matrices of learnable parameters. As we now explain, the calculation of $\widetilde{\boldsymbol{b}}_t$ is carried out by the unbinding and binding operations of TPRs (Eq. 1).

## 3.1 UNBINDING OPERATION

Consider a set of hypothesised unbinding vectors $\boldsymbol{U} = [\boldsymbol{u}_1, \boldsymbol{u}_2, ..., \boldsymbol{u}_N] \in \mathbb{R}^{d \times N}$. At time step $t$, these can be used to unbind fillers $\boldsymbol{f}_{b,t}$ from the previous binding complex $\boldsymbol{b}_{t-1}$ using Eq. 1. We posit two matrices $\boldsymbol{V}_x \in \mathbb{R}^{d \times d'}$ and $\boldsymbol{V}_b \in \mathbb{R}^{d \times d}$ that transform the current input $\boldsymbol{x}_t \in \mathbb{R}^{d'}$ and the binding complex $\boldsymbol{b}_{t-1}$ into the binding space $\mathbb{R}^d$ yielding fillers $\boldsymbol{f}_{x,t}$ and $\boldsymbol{f}_{b,t}$ :

$$\boldsymbol{f}_{b,t} = \boldsymbol{U}^\top \boldsymbol{V}_b \boldsymbol{b}_{t-1} \in \mathbb{R}^N, \qquad \boldsymbol{f}_{x,t} = \boldsymbol{U}^\top \boldsymbol{V}_x \boldsymbol{x}_t \in \mathbb{R}^N. \tag{3}$$

Normalisation is applied for stable learning by gradient descent, and the specific technique (Yang et al., 2018) we adopted here also enforces sparsity on both $\boldsymbol{f}_{b,t}$ and $\boldsymbol{f}_{x,t}$:

$$(\boldsymbol{f}_t)_n = (\widetilde{\boldsymbol{f}}_t)_n^2 / \sum_{m=1}^N (\widetilde{\boldsymbol{f}}_t)_m^2 \tag{4}$$

$$\text{where } (\widetilde{\boldsymbol{f}}_t)_n = \max(0, (\boldsymbol{f}_{b,t})_n + b_b) + \max(0, (\boldsymbol{f}_{x,t})_n + b_x) \tag{5}$$

in which $b_b$ and $b_x$ are two scalar parameters for stable learning, and $(\cdot)_n$ refers to the $n$-th entry of the vector in the parentheses.

## 3.2 BINDING OPERATION

Given a hypothesised set of binding role vectors $\boldsymbol{R} = [\boldsymbol{r}_1, \boldsymbol{r}_2, ..., \boldsymbol{r}_N] \in \mathbb{R}^{d \times N}$, applying the binding operation in Eq. 1 to the fillers $\boldsymbol{f}_t$ at time $t$ gives the candidate update $\widetilde{\boldsymbol{b}}_t$ for the binding complex,

$$\widetilde{\boldsymbol{b}}_t = \boldsymbol{R} \boldsymbol{f}_t. \tag{6}$$

The gating mechanism controls the weighted sum of the candidate vector $\widetilde{\boldsymbol{b}}_t$ and the previous binding complex $\boldsymbol{b}_{t-1}$ to produce a binding complex $\boldsymbol{b}_t$ at the current time step, as given by Eq. 2.

## 3.3 UNBINDING AND BINDING ROLE VECTORS

In the TPRU, there is a matrix of role vectors $\boldsymbol{R}$ used for the binding operation and a matrix of unbinding vectors $\boldsymbol{U}$ used for the unbinding operation. To control the number of parameters in our TPRU, rendering it independent of $N$, instead of directly learning two sets of vectors, a set of $N$ $d$-dimensional random vectors $\boldsymbol{E}$ is sampled from a normal distribution, and two linear projections $\boldsymbol{W}_r, \boldsymbol{W}_u \in \mathbb{R}^{d \times d}$ are learnt to transform $\boldsymbol{E}$ to $\boldsymbol{U} = \boldsymbol{W}_u \boldsymbol{E}$ and $\boldsymbol{R} = \boldsymbol{W}_r \boldsymbol{E}$.

Therefore, in total, our proposed TPRU has six learnable matrices: $\boldsymbol{W}_u, \boldsymbol{W}_r, \boldsymbol{V}_b, \boldsymbol{W}_b \in \mathbb{R}^{d \times d}$ and $\boldsymbol{V}_x, \boldsymbol{W}_x \in \mathbb{R}^{d \times d'}$. The number of parameter matrices the same as that of a GRU and less than that of an LSTM, and, when $d' < d$, the number of learnable parameters is less than that of a GRU.

## 4 GRADIENT ANALYSIS AND COMPLEXITY

This section analyses the derivatives in our proposed TPRU, and aims to justify the architecture design. For simplicity, we omit the derivatives w.r.t. bias terms in our derivations as biases can be absorbed into the weight matrices by adding an extra component 1 to the input.

As known, the gradient of an RNN cell is accumulated in time $\frac{\partial \boldsymbol{b}_T}{\partial \boldsymbol{b}_1} = \prod_{t=1}^{T-1} \frac{\partial \boldsymbol{b}_{t+1}}{\partial \boldsymbol{b}_t}$, therefore, studying the properties of the derivative of the output at the current time step $\boldsymbol{b}_t$ (Eq. 2) w.r.t. the output

of the previous time step $\boldsymbol{b}_{t-1}$ is important for understanding the behaviour of a recurrent unit. For our TPRU, the derivative is shown below:

$$\frac{\partial \boldsymbol{b}_t}{\partial \boldsymbol{b}_{t-1}} = \frac{\partial \boldsymbol{b}_t}{\partial \boldsymbol{g}_t}\frac{\partial \boldsymbol{g}_t}{\partial \boldsymbol{b}_{t-1}} + \frac{\partial \boldsymbol{b}_t}{\partial \widetilde{\boldsymbol{b}}_t}\frac{\partial \widetilde{\boldsymbol{b}}_t}{\partial \boldsymbol{b}_{t-1}} + \frac{\partial ((1 - \boldsymbol{g}_t) \circ \boldsymbol{b}_{t-1})}{\partial \boldsymbol{b}_{t-1}} = \boldsymbol{\Lambda}_1 \boldsymbol{W}_b + \boldsymbol{\Lambda}_2\frac{\partial \widetilde{\boldsymbol{b}}_t}{\partial \boldsymbol{b}_{t-1}} + \boldsymbol{\Lambda}_3 \tag{7}$$

where $\boldsymbol{\Lambda}_1$, $\boldsymbol{\Lambda}_2$ and $\boldsymbol{\Lambda}_3$ are diagonal matrices in $\mathbb{R}^{d\times d}$, in which $(\boldsymbol{\Lambda}_1)_{ii} = (\widetilde{\boldsymbol{b}}_t)_i (\boldsymbol{g}_t)_i (1 - (\boldsymbol{g}_t)_i)$, $(\boldsymbol{\Lambda}_2)_{ii} = (\boldsymbol{g}_t)_i$ and $(\boldsymbol{\Lambda}_3)_{ii} = (1 - (\boldsymbol{g}_t)_i)$. It is clear that the gating mechanism smooths the gradient flow as it jitters the Jacobian matrix by adding small values to the diagonal terms, and from a dynamic system perspective, it helps stabilise the learning system (Moya-Cessa & Soto-Eguibar, 2011). Now we focus on the following derivative which carries the main information of our proposed TPRU.

$$\frac{\partial \widetilde{\boldsymbol{b}}_t}{\partial \boldsymbol{b}_{t-1}} = \frac{\partial \widetilde{\boldsymbol{b}}_t}{\partial \boldsymbol{f}_t}\frac{\partial \boldsymbol{f}_t}{\partial \boldsymbol{f}_{b,t}}\frac{\partial \boldsymbol{f}_{b,t}}{\partial \boldsymbol{b}_{t-1}} = (\boldsymbol{W}_r \boldsymbol{E})(\boldsymbol{J}_t \boldsymbol{N}_t \boldsymbol{S}_{b,t})(\boldsymbol{E}^\top \boldsymbol{W}_u^\top \boldsymbol{V}_b) \tag{8}$$

where $\boldsymbol{N}_t, \boldsymbol{S}_{b,t} \in \mathbb{R}^{N\times N}$ are diagonal matrices, in which $(\boldsymbol{N}_t)_{nn} = 2/(\widetilde{\boldsymbol{f}}_t)_n$ and $(\boldsymbol{S}_{b,t})_{nn} = \mathbb{1}_{(\boldsymbol{f}_{b,t})_n > 0}$, and $(\boldsymbol{J}_t)_{mn} = (\boldsymbol{f}_{b,t})_m(\delta_{mn} - (\boldsymbol{f}_{b,t})_n)$. Appendix A shows that $\boldsymbol{J}_t$ is a positive semi-definite matrix, and its eigenvalues are all smaller than $\max_n(\boldsymbol{f}_{b,t})_n \leq 1$.

There are three crucial design choices, including (1) learnable parameter matrices $\boldsymbol{W}_r$ and $\boldsymbol{W}_u$ to transform the basis matrix $\boldsymbol{E}$ separately, (2) learnable linear transformation $\boldsymbol{V}_b$ on the binding complex at each time step $\boldsymbol{b}_t$, and (3) the relu-squared normalisation in Eq. 4. The following analysis shows that certain modifications could lead to issues during learning, and helps understand the necessity of each component.

**Modification I.** $\boldsymbol{W}_r = \boldsymbol{W}_u$, $\boldsymbol{V}_b = \boldsymbol{I}_d$ and softmax normalisation in Eq. 4.

The derivative of the softmax normalisation of outputs w.r.t. the inputs is simply $\boldsymbol{J}_t$ as $\boldsymbol{S}_{b,t} = \boldsymbol{N}_t = \boldsymbol{I}_N$.[1] Given the conditions, the derivative can be simplified to:

$$\partial \widetilde{\boldsymbol{b}}_t / \partial \boldsymbol{b}_{t-1} = \boldsymbol{W}_r \boldsymbol{E} \boldsymbol{J}_t \boldsymbol{N}_t \boldsymbol{S}_{b,t} \boldsymbol{E}^\top \boldsymbol{W}_u^\top \boldsymbol{V}_b = \boldsymbol{W}_r \boldsymbol{E} \boldsymbol{J}_t \boldsymbol{E}^\top \boldsymbol{W}_r^\top \succeq 0 \tag{9}$$

In this case, as the derivative gets multipled several times by BPTT, the gradient w.r.t. each parameter matrix either explodes or vanishes depending on the spectrum of $\boldsymbol{W}_r \boldsymbol{E}$. Also, a careful initialisation is required otherwise the gradient could overflow or underflow after the first iteration.

**Modification II.** $\boldsymbol{W}_r = \boldsymbol{W}_u$, and softmax normalisation.

Here we allow learnable parameter matrices $\boldsymbol{V}_b$ to transform the binding complex from the previous time step $\boldsymbol{b}_{t-1}$, thus, the derivative of $\widetilde{\boldsymbol{b}}_t$ w.r.t. $\boldsymbol{b}_{t-1}$ and that to $i$-th row of $\boldsymbol{V}_b$ are shown below:

$$\partial \widetilde{\boldsymbol{b}}_t / \partial \boldsymbol{b}_{t-1} = \boldsymbol{W}_r \boldsymbol{E} \boldsymbol{J}_t \boldsymbol{E}^\top \boldsymbol{W}_r^\top \boldsymbol{V}_b \qquad \partial \widetilde{\boldsymbol{b}}_t / \partial (\boldsymbol{V}_b)_{i\cdot} = (\boldsymbol{W}_r \boldsymbol{E} \boldsymbol{J}_t \boldsymbol{E}^\top \boldsymbol{W}_r^\top)_{\cdot i} \boldsymbol{b}_{t-1} \tag{10}$$

Since $\boldsymbol{W}_r \boldsymbol{E} \boldsymbol{J}_t \boldsymbol{E}^\top \boldsymbol{W}_r^\top \succeq 0$, $\partial \widetilde{\boldsymbol{b}}_t / \partial (\boldsymbol{V}_b)_{i\cdot}$ always has a non-negative inner product with $\boldsymbol{b}_{t-1}$, which means that the update of $\boldsymbol{V}_b$ always contains a positive portion of $\boldsymbol{b}_{t-1}$. The same situation also applies when updating $\boldsymbol{V}_x$ as it keeps a positive portion of $\boldsymbol{x}_t$ as well. It limits the learning system from eliminating accumulated useless information. Although it is still possible that, after backpropagation through several time steps, the positive portion of $\boldsymbol{b}_{t-1}$ could be suppressed, we set $\boldsymbol{W}_r$ and $\boldsymbol{W}_u$ to be independent matrices so that the above constraint isn't imposed.

**Modification III.** Softmax normalisation.

The only modification here is to replace the normalisation step in Eq. 4 in our TPRU to a softmax function that also normalises $\widetilde{\boldsymbol{f}}_t$ to form a distribution $\boldsymbol{f}_t$. At each time step, the derivative of $\widetilde{\boldsymbol{b}}_t$ w.r.t. any parameter matrix before the normalisation step contains $\partial \boldsymbol{f}_t / \partial \widetilde{\boldsymbol{f}}_t$, and in the softmax normalisation, only the relative difference, $(\widetilde{\boldsymbol{f}}_t)_n - \max_m(\widetilde{\boldsymbol{f}}_t)_m, \forall n \in \{1, 2, ..., N\}$, matters.

Therefore, there is no explicit regularisation in the derivative of $\partial \widetilde{\boldsymbol{b}}_t / \partial \boldsymbol{b}_{t-1} = \boldsymbol{W}_r \boldsymbol{E} \boldsymbol{J}_t \boldsymbol{E}^\top \boldsymbol{W}_u^\top \boldsymbol{V}_b$ to penalise extreme values in $\widetilde{\boldsymbol{f}}_t$, which could lead to gradients with large norms and make the learning process unstable. This may also explain why, in the transformer module (Vaswani et al., 2017), layer normalisation (Ba et al., 2016) is necessary, as it scales the gradient by the variance of the input.

We don't rule out the possibility that there exists a suitable initialisation which gives stable learning and decent performance, but the normalisation step we adopted from prior work (Yang et al., 2018) avoids heavy tuning of initialisations for stable learning.

---

[1] $I_N$ is an identity matrix in $\mathbb{R}^{N\times N}$.

**Ours.** Here, we recall that the derivative of $\partial \widetilde{\boldsymbol{b}}_t / \partial \boldsymbol{b}_{t-1}$ is in Eq. 8. The term $\boldsymbol{N}_t$ scales the derivative by the magnitude of the input $\widetilde{\boldsymbol{f}}_t$ to Eq. 4, which regularises the norm of the gradient to each parameter matrix. Specifically, the following derivative indicates the update of $i$-th row in $\boldsymbol{W}_u$:

$$\frac{\partial \widetilde{\boldsymbol{b}}_t}{\partial (\boldsymbol{W}_u)_{i\cdot}} = \frac{\partial \widetilde{\boldsymbol{b}}_t}{\partial \boldsymbol{f}_{b,t}} \frac{\partial \boldsymbol{f}_{b,t}}{\partial (\boldsymbol{W}_u)_{i\cdot}} + \frac{\partial \widetilde{\boldsymbol{b}}_t}{\partial \boldsymbol{f}_{x,t}} \frac{\partial \boldsymbol{f}_{x,t}}{\partial (\boldsymbol{W}_u)_{i\cdot}} = \boldsymbol{W}_r \boldsymbol{E} \boldsymbol{J}_t \boldsymbol{N}_t \left( \boldsymbol{S}_{b,t} (\boldsymbol{V}_b \boldsymbol{b}_{t-1})_i + \boldsymbol{S}_{x,t} (\boldsymbol{V}_x \boldsymbol{x}_t)_i \right) \boldsymbol{E}^\top \quad (11)$$

where $\boldsymbol{N}_t$ normalises the contribution from $\boldsymbol{f}_{b,t}$ and $\boldsymbol{f}_{x,t}$, and $\boldsymbol{S}_{b,t}$ and $\boldsymbol{S}_{x,t}$ select positive contributions only, thus the normalisation step in Eq. 4 helps the gradients w.r.t. parameter matrices to stay stable. Eq. 11 also shows that the update of each row in $\boldsymbol{W}_u$ depends on a sparse linear transformation of $\boldsymbol{W}_r$, which could potentially avoid overfitting as not all dimensions in the representation space are utilised and our TPRU learns to discover the representation space.

Given the derivative above, it is easy to see that the presampled matrix $\boldsymbol{E}$ acts similarly as the preconditioning matrix in solving large linear systems, and with samples from a normal distribution or zero-centred uniform distribution, it smooths the spectrum of the Jacobian matrix and stabilizes the learning system.

**Time Complexity in Forward Computation**

Table 1 presents the time complexity of calculating each of our proposed TPRU, LSTM and GRU for a single time step. In the calculation, we only kept the terms that involve second order of variables. As shown, the time complexity of TPRU is in between LSTM and GRU when the number of role vectors $N$ is less than the dimension of the recurrent unit $d$. However, the number of parameters in our TPRU is fewer than in an LSTM, and comparable to a GRU.

Table 1: **Complexity**

| Model | Time Complexity | # Params |
|-------|-----------------|----------|
| TPRU  | $O(2Nd + 6d^2)$ | $4d^2 + 2dd'$ |
| LSTM  | $O(8d^2)$       | $4d^2 + 4dd'$ |
| GRU   | $O(6d^2)$       | $3d^2 + 3dd'$ |

In addition, as concerns nonlinearities, our TPRU contains $\max(x, 0)^2$ only $N$ times, and sigmoid functions only $d$ times; this is fewer than sigmoid/tanh squashing functions appearing $5d$ times in an LSTM and $3d$ times in a GRU. The nonlinear operations are indeed reduced in our TPRU, and the following sections show that our TPRU still provides strong performance on evaluation tasks.

## 5 TASKS AND TRAINING DETAILS

Three entailment tasks, including an abstract logical entailment task (Evans et al., 2018) and two relatively realistic natural language entailment tasks, including Multi-genre Natural Language Inference (MNLI, Williams et al. (2017)) and Question Natural Language Inference (QNLI, Wang et al. (2018)), as well as a language modelling task on a large corpus and a small one, (WikiText-103 and WikiText-2 (Merity et al., 2018)), are considered to demonstrate the learning ability of our proposed TPRU. Each of the three entailment tasks provides pairs of samples, and for each pair, the model needs to tell whether the first (the premise) entails the second (the hypothesis). Language modelling is used to demonstrate the ability of our proposed model on sequential modelling.

Experiments are conducted in PyTorch (Paszke et al., 2017) with the Adam optimiser (Kingma & Ba, 2014) and gradient clipping (Pascanu et al., 2013). Reported results are averaged from the results of three random initialisations. Our proposed TPRU has only one input gate and also has similar number of parameters, so the GRU is the most relevant comparison partner. LSTM is also included as it is widely-used as well and it generally yields better performance than GRU does. For each model, hyperparameters are searched based on the small selected validation set on each task, including dropout rates, learning rate and batch size. For entailment tasks, we tested both plain and BiDAF architectures (Seo et al., 2017). On both NLI tasks, ELMo (Peters et al., 2018) encoding is applied for embedding words into contextualised vectors. More details about the tasks and the training details can be found in Sections B and C in the Appendix.

## 6 DISCUSSION

On the Logical Entailment task in Table 2, when it comes to larger dimensionality ($N = 512$), our TPRU appears to be more stable than GRU during learning as we observed that the GRU with the

Table 2: **Accuracy of each model on the propositional Logical Entailment task.** *Overall, our proposed TPRU outperforms its closest comparison counterpart GRU, and provides comparable performance to the LSTM.* The value besides 'TPRU' indicates the number of role vectors. Results of BiDAF models are parenthesised. 'Mean $2^{\#\text{Vars}}$' is the mean number of worlds of proposition-values; here and below, **bold** indicates the best result among all models with the same architecture.

| model | valid | | | test | | | # params |
|---|---|---|---|---|---|---|---|
| | | easy | hard | big | massive | exam | |
| Mean $2^{\#\text{Vars}}$ | 75.7 | 81.0 | 184.4 | 3310.8 | 848,570.0 | 5.8 | |
| **Plain (BiDAF) Architecture** - dim 64 | | | | | | | |
| GRU | 84.3 (**91.3**) | 68.2 (76.0) | 84.3 (**91.3**) | 77.0 (80.0) | 57.4 (**70.4**) | 66.8 (74.3) | 49.1k (172.4k) |
| TPRU - 512 | **88.6** (90.9) | 73.1 (**77.18**) | **88.4** (90.6) | **79.0** (**82.0**) | 62.0 (69.9) | 71.8 (**74.9**) | 98.3k (213.0k) |
| LSTM | **88.6** (89.1) | **76.3** (75.4) | **88.4** (89.1) | **79.0** (75.0) | **64.7** (66.7) | **76.4** (**74.9**) | 131.1k (327.7k) |

BiDAF architecture failed to converge in two out of three trials. On certain test sets, our (singly-gated) TPRU slightly underperforms the LSTM, possibly due to the three gates (and more parameters) of the LSTM: future research could incorporate more sophisticated gating mechanisms into the TPRU.

On the MNLI tasks in Table 3, our proposed TPRU consistently outperforms the LSTM when a large number of role vectors is deployed. We omit the results from GRUs as they are worse than LSTMs and often unstable in learning. Unexpectedly, all models perform better on the mismatched dev set than on the matched one, perhaps because it happens to be easier.

Table 3: **Results on MNLI and QNLI.** *Our TPRU outperforms LSTM.* The number of role vectors in each of our models is indicated by the value beside 'TPRU'. As shown, our proposed TPRU overall outperforms LSTM on two tasks.

| model | MNLI - dev | | QNLI - dev | # params |
|---|---|---|---|---|
| | matched | mismatched | | |
| **Plain (BiDAF) Architecture** - dim 512 | | | | |
| TPRU - 256 | 74.2 (77.7) | 74.9 (78.0) | 78.2 (**81.2**) | |
| TPRU - 512 | **74.8** (**78.1**) | **75.6** (77.8) | **78.8** (80.1) | |
| TPRU - 1024 | 74.4 (77.7) | 74.8 (**78.5**) | **78.8** (80.8) | 8.4m (16.8m) |
| LSTM | 73.4 (76.5) | 73.9 (77.0) | 77.2 (79.1) | 12.6m (27.3m) |
| **Plain (BiDAF) Architecture** - dim 1500 | | | | |
| TPRU - 512 | 74.5 (77.9) | 75.4 (**78.8**) | 79.3 (**82.5**) | |
| TPRU - 1024 | 74.8 (77.9) | **76.1** (78.3) | **80.4** (82.2) | |
| TPRU - 2048 | **75.1** (**78.7**) | 75.2 (**78.8**) | 80.0 (**82.5**) | 48.3m (108.4m) |
| LSTM | 74.8 (77.1) | 75.7 (77.7) | 78.8 (81.2) | 60.6m (162.9m) |
| SOTA (ALICE ensemble) | 88.2 | 87.9 | 95.7 | |

On the language modelling task on WikiText-103 and WikiText-2, our TPRU provides similar perplexity on the test set to the LSTM, as shown in Table 4. The learning plot in Figure 4 shows that our TPRU learns faster on the training set than the GRU and the LSTM. The learning plot and the performance table both serve as evidential support for the gradient analysis of our model.

## 6.1 POLYSEMY

In TPR, multiple senses of a single polysemous word can be encoded by binding the same symbol that represents the polysemous word to different role vectors $\{r_i\}$. TPR is composing a distributed representation that keeps multiple senses of a polysemous word and recovering each sense with no information loss if a large number of dimensions is available. In our case, the magnitude of each entry in $f_t$ determines the contribution of each role vector to the update candidate $\widetilde{b}_t$ (Eq. 6).

Table 4: **Perplexity of language modelling on WikiText-2 and WikiText-103.** (Lower is better.) **(left):** Our TPRU with 1024 role vectors gives similar perplexity score as the widely used LSTM does. **(right):** Negative Log-likelihood on training set of language modelling task on WikiText-103. (The plotted models are trained without dropout layers.) Our TPRU converges faster on the training set than the GRU and LSTM. As the GRU doesn't generalise very well on the dev test, its results are omitted from the table.

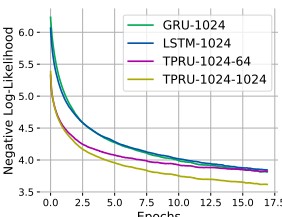

| model | WikiText-103 | WikiText-2 | #Params |
|-------|--------------|------------|---------|
| **Two-layer Architecture** - dim 1024 | | | |
| TPRU | 41.4 | 76.8 | 12.6m |
| LSTM | 42.3 | 76.0 | 16.8m |
| SOTA | 16.4 (Krause et al., 2019) | 39.1 (Gong et al., 2018) | 257m / 35m |

We train our TPRU with a BiDAF architecture on the MNLI dataset, and word vectors are obtained from fastText (Bojanowski et al., 2017) as the input to our model so that the vector representation of a polysemous word stays the same even when it is presented in different contexts. After learning, our model provides an interface for us to check whether a specific role vector correlates to a specific sense of a polysemous word in different phrases in the dev set by computing $\beta_t = \arg\max_n(\boldsymbol{f}_t)_n$.

For example, a common polysemous word bank has two senses when it is used as a noun. In the dev set of MNLI, the only case when bank is used in the river-context is in the phrase "*bank fishing*", and the rest belong to the other sense in the context of economy. We observed that $\beta$ is the same when bank occurs in "*bank fishing*" and we denote it as $\beta_{\text{river}}$, thus, $P(\beta = \beta_{\text{river}}|\underline{\text{bank}}$ in the river-context$) = 1$. It is also important to see if the specific $\beta_{\text{river}}$ is dedicated to the word bank in the river context. After examining the dev set, we have $P(\underline{\text{bank}}$ in the river-context$|\beta = \beta_{\text{river}}) = P(\underline{\text{bank}}$ in "*the west bank*"$|\beta = \beta_{\text{river}}) = 0.5$; clearly, the sense of "*bank*" relevant to the phrase "*the west bank*" is the one related to rivers, not finances.

## 6.2 POS TAGGING

We calculated the pointwise mutual information (PMI) between the following two terms: 1) The maximally selected role vector at each time step $\beta_t = \arg\max_n(\boldsymbol{f}_t)_n$ and 2) The part-of-speech tag (POS Tag) provided by a pretrained tagger.[2] For a given POS tag, the PMIs of role vectors are sorted in a descending order, and Table 5 presents the top two PMI values for those POS tags for which these two values differ by at least $0.5$: this includes 7 out of 17 fine-grained POS tags. The idea is that if the top PMI value is sufficiently greater than the next one, and hence all the rest, then it is justified to say that each POS tag in the table shown below has a role 'dedicated' to it.

Table 5: **PMI between POS Tags and Maximally Selected Role Vectors.**

| POS Tags | ADV | DET | INTJ | NOUN | PART | PROPN | VERB |
|----------|-----|-----|------|------|------|-------|------|
| Top 2 PMI | 7.37 / 5.83 | 6.06 / 5.41 | 7.13 / 6.46 | 7.23 / 5.84 | 7.54 / 6.91 | 7.27 / 6.47 | 7.22 / 6.28 |

As the model is only trained on MNLI to predict the entailment relationship between a given pair of sentences without access to POS tags, it is interesting that some role vectors become associated with specific POS tags. Since POS tags carry structural information of a given sentence, and since the roles learnt by the TPRU are informative about the POS tags, we have evidence that the roles that explicitly define the structure of the model's representations are capturing some of the distinctions between words that are believed to be fundamental to linguistic structure.

As pointed out in prior work, models trained on MNLI focus on words in lexical (non-functional) categories in determining the entailment relationship between a sentence pair; here we see that POS tags of lexical words, including NOUN, PROPN, VERB, etc., have a larger gap between the top 2 highest PMI values as they are apparently emphasised by the learning signal. This finding suggests

---

[2]https://spacy.io/api/doc

that our learnt model might help to identify the biases in datasets — which could be helpful in improving fairness and inclusiveness.

The above two empirical observations point out that our proposed TPRU inherits the interpretability of the original TPRs, and provides a certain level of human-readability. It provides a simple interface for us to analyse the learnt model through the explicit binding and unbinding operations.

### 6.3 Effect of Increasing the Number of Role Vectors

In the original TPRs, roles are typically filled by at most a single symbol, so the total number of distinct roles indicates the number of symbols that can be used in the representation of a single data item. Since each symbol is capable of representing a specific substructure of the input data, increasing the number of role vectors eventually leads to more highly structured representations if there is no limit on the dimensionality of role vectors.

Experiments are conducted to show the effect of increasing the number of role vectors on the performance on both Logical Entailment and MNLI. As shown in Table 3, adding more role vectors into our proposed TPRU gradually improves the performance on the two entailment tasks.

Figure 1 presents the learning curves, including training loss and accuracy, of our proposed TPRU with varying number of role vectors on the two entailment tasks. As shown in the graphs, incorporating more role vectors leads to not only better performance, but also faster convergence during training. The observation is consistent on both the Logical Entailment and MNLI tasks.

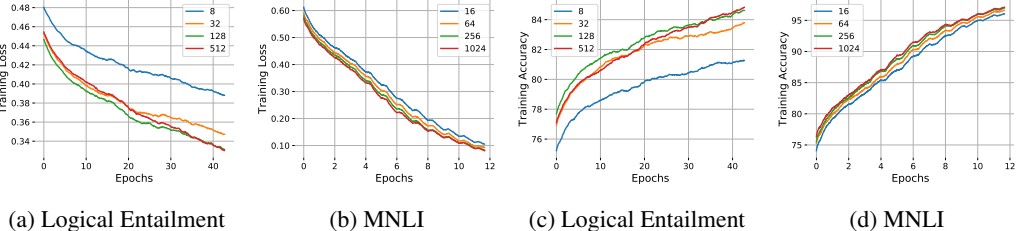

| (a) Logical Entailment | (b) MNLI | (c) Logical Entailment | (d) MNLI |

Figure 1: **Training Plots of our TPRU with different number of role vectors**. (The plotted models are trained without dropout layers.) In general, our proposed recurrent unit converges faster and leads to better results on both tasks when more role vectors are available, but the total number of parameters remains the same. (Best viewed in colour.)

Interestingly, on the Logical Entailment task, the performance of our proposed TPRU with only eight role vectors matches that of a GRU with the same dimensionality. The number of role vectors required in our TPRU to approximately match the performance of a GRU and an LSTM on the MNLI task is much less than the dimension of the hidden states, which suggests that the distributed vector representations learnt in both the LSTM and GRU are highly redundant and can be reduced to fewer dimensions; this also implies that the LSTM and GRU are not able to extensively exploit the representation space even when the dimensionality of the model is larger than that required for solving the task. Meanwhile, our proposed TPRU explores sparse weighted combinations of role vectors in Eq. 4, and it helps the model to specialise individual role vectors for different purposes, which eventually leads to a more effective exploitation of the representation space.

## 7 Conclusion

We have proposed a simple recurrent unit that leverages the explicit binding and unbinding operations in previously proposed TPRs in a reduced fashion, therefore it is named the TPRU. The explicit execution arms the proposed model with the ability to learn decomposable representations, and by sharing the role vectors across time steps and sequences, our TPRU provides research an intuitive interface through $f_t$ to study the information encoded in the learnt distributed vector representations.

Since recurrent neural networks are prone to suffer from the gradient vanishing and exploding issues, we conduct a thorough analysis of the derivatives and the gradient update of our proposed TPRU on

each time step. The conducted analysis supports our decisions in designing TPRU, and it shows that it encourages relatively more stable gradient flow through BPTT. The time complexity for executing one step of our TPRU is lower than that of LSTM, and it has the same or a smaller number of parameters than the GRU does, which justifies the simplicity of our recurrent unit.

The empirical evaluation on five learning tasks in three different categories demonstrate the effectiveness of our TPRU as it performs on a par with or better than the LSTM. The study on increasing the number of role vectors and plots on the loss curves during learning shows that our model converges faster than the GRU and the LSTM in general and it is a result of the stable gradient flow through multiple time steps. In addition, our proposed TPRU enables a faster convergence rate and better performance by increasing the number of role vectors.

A study on correlating the role vectors with linguistic properties of languages shows that our TPRU inherits the strong interpretability of TPRs. In the toy example involving a polysemous word bank, our model can differentiate multiple senses of the word, and can assign a similar sense, even in a named entity, to its closely related one. Also, the PMI values between POS tags and maximally selected role vectors imply that our TPRU is capable of discovering syntactically-structured linguistic information.

Given the current pressing need for interpretable machine learning and learning decomposable representations, our proposed TPRU can be regarded as a more light-weight and human-readable alternative to the widely-used LSTM or GRU. Future work will focus on incorporating full TPRs into the design of an RNN unit and into the Transformer module.

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

APPENDIX

## A    PROOF

**1. $J_t$ is a PSD matrix.** For any non-zero vector $v$, we have

$$v^\top J_t v = \sum_{n=1}^{N} (f_{b,t})_n v_n^2 - (\sum_{n=1}^{N} (f_{b,t})_n v_n)^2 \tag{12}$$

$$\geq (\sum_{n=1}^{N} (f_{b,t})_n v_n)^2 - (\sum_{n=1}^{N} (f_{b,t})_n v_n)^2 = 0 \qquad \text{(Jensen's Inequality)} \tag{13}$$

Therefore, $J_t$ is a PSD matrix.

## B    DETAILS ABOUT TASKS

Three entailment tasks, including an abstract logical entailment task and two relatively realistic natural language entailment tasks, as well as a language modelling task on a large corpus and a small one, are considered to demonstrate that the learning ability of our proposed TPRU. Each of the three entailment tasks provides pairs of samples, and for each pair, the model needs to tell whether the first (the premise) entails the second (the hypothesis). Language modelling is used to demonstrate the ability of our proposed model on sequential modelling.

As our goal is to learn structured vector representations, the proposed TPRU serves as an encoding function, which learns to process a proposition or sentence one token at a time, and produce a vector representation. During learning, two vector representations are produced by the same recurrent unit given a pair of samples, then a simple feature engineering method (e.g. concatenation of the two representations) is applied to form an input vector for subsequent classification. In general, with a simple classifier, e.g. a linear classifier or a multi-layer perceptron with a single hidden layer, the learning process forces the encoding function to produce high-quality representations of samples, either propositions or sentences; better representations enable stronger performance.

### B.1    LOGICAL ENTAILMENT (PROPOSITIONAL LOGIC)

In propositional logic, for a pair of propositions, $A$ and $B$, the value of $A \vDash B$ is independent of the identities of the shared variables between $A$ and $B$, and is dependent only on the structure of the expression and the connectives in each subexpression. Because of $\alpha$-equivalence, $p \wedge q \vDash q$ holds no matter how we replace variable $p$ or $q$ with any other propositional variables.[3] Logical entailment is

---

[3]For example, $p \wedge q \vDash q \Leftrightarrow a \wedge b \vDash a$.

thus a good test-bed for evaluating a model's ability to carry out abstract, highly structure-sensitive reasoning (Evans et al., 2018).

Theoretically, it is possible to construct a truth table that contains as rows/worlds all possible combinations of values of variables in both propositions $A$ and $B$; the value of $A \vDash B$ can be checked by going through every entry in each row. As the logical entailment task emphasises reasoning on connectives, it requires the learnt distributed vector representations to encode the structure of any given proposition to excel at the task.

The Logical Entailment dataset[4] has balanced positive ($A \vDash B$) and negative ($A \nvDash B$) classes and a representative validation set. Five test sets are provided to evaluate the generalisation ability at different difficulty levels: some test sets have significantly more variables and operators than the training and validation sets (see Table 2).

## B.2 Natural Language Inference

Natural Language Inference — determining whether a premise sentence entails a hypothesis sentence — combines the structure-dependence of logical entailment with the NLP tasks of inferring word meaning in context as well as the hierarchical relations among sentence constituents. Two datasets are considered, which are Multi-genre NLI and QNLI (Williams et al., 2017; Wang et al., 2018). Compared to logical entailment, the inference and reasoning in NLI also relies on the identities of words in sentences in addition to their structure. More importantly, the ambiguity and polysemy of language lead to the impossibility of creating a truth table that lists all cases. Therefore, NLI is an intrinsically hard task.

The Multi-genre Natural Language Inference (MNLI) dataset (Williams et al., 2017) collects sentence pairs in ten genres; only five genres are available in the training set, while all ten genres are presented in the development set. QNLI (Wang et al., 2018) is collected from Stanford Question Answering dataset (SQuAD) (Rajpurkar et al., 2016), on which a model needs to predict whether the context sentence contains the answer to the question sentence or not. The relation between sentences in a pair must be classified as Entailment, Neutral or Contradiction.

The performance of a model on the five 'mismatched' genres that exist in the development but not the training set reflects how well the structure encoded in distributed vector representations of sentences learnt from genres seen in training generalises to sentence pairs in unseen genres. As the nature of NLI tasks requires inferring both word meaning and structure of constituents in a given sentence, supervised training signals from labelled datasets enforce an encoding function to analyse meaning and structure at the same time during learning, which eventually forces distributed vector representations of sentences produced from the learnt encoding function to be structured. To the extent that the inferences required in an NLI task are demanding of proper analysis of the logical structure implicit in sentence, an inductive bias that enhances the ability to learn to encode the structures of sentences should enhance success on the MNLI task.

## B.3 Language Modelling

Language modelling is a fundamental task to test generalisation and sequential modelling. Recent research has shown that models trained on language modelling tasks yield representations that can transfer well to other natural language tasks (Peters et al., 2018; Radford et al., 2018; Devlin et al., 2018). We follow the autoregressive setting of decomposing the joint probability on tokens into a product of conditional probabilities on generating the current token given tokens in the left context.

$$P(w_1, w_2, ..., w_L) = \prod_{l=1}^{L} P(w_l | w_{l-1}, w_{l-2}, ..., w_1). \tag{14}$$

This requires the learnt representation to capture the potentially highly structured information accumulated prior to the current time step in order to make a confident prediction of the current token. A proper inductive bias towards constructing structured representations should enhance performance on the task. Two corpora used in our experiment are WikiText-103 and WikiText-2 (Merity et al., 2018), in which one contains 103 million tokens in the training set while the other only has 2 million. The results will tell us how well our proposed TPRU generalises across different training scales.

---

[4]https://github.com/deepmind/logical-entailment-dataset

## C    DETAILS ABOUT ARCHITECTURE

### C.1    PLAIN ARCHITECTURE - INFERENCE:

For both the Logical Entailment task and the MNLI task, we train two-layer recurrent networks built using our proposed TPRU, networks built using its direct comparison counterpart, the GRU, as well as LSTMs. A global max-pooling over time is applied to the hidden states produced by each recurrent unit at all time steps to generate the vector representation for a given logical proposition or sentence. Given a pair of generated representations $u$ and $v$, a vector is constructed to represent the difference between two vectors $[u; v; |u - v|; u \circ v]$, where $u \circ v$ is the Hadamard product and $|u - v|$ is the absolute difference, and the vector is fed into a multi-layer perceptron which has only one hidden layer with ReLU activation function. The feature engineering and the choice of classifier are suggested by prior work (Seo et al., 2017; Wang et al., 2018).

Symbolic Vocabulary Permutation (Evans et al., 2018) is applied as data augmentation during learning on the logical entailment task: exploiting $\alpha$-equivalence, this systematically replaces the variables in a proposition pair with randomly sampled alternative variables, as only connectives matter on this task. Table 2 presents the results. For the MNLI task, the Stanford Natural Language Inference (SNLI) dataset (Bowman et al., 2015) is included as additional training data as recommended (Wang et al., 2018), and ELMo (Peters et al., 2018) is applied for producing word vectors. The best model is chosen according to the averaged classification accuracy on the five matched and five mismatched dev sets. The results are presented in Table 3. SOTA models come from Alibaba DAMO NLP Group, and the results are available on the GLUE leaderboard. [5]

### C.2    BiDAF ARCHITECTURE - INFERENCE:

Bi-directional Attention Flow (**BiDAF**) (Seo et al., 2017) has been successfully applied to various NLP tasks (Seo et al., 2017; Chen et al., 2017), and provides strong performance on NLI tasks (Wang et al., 2018). The BiDAF architecture contains a layer for encoding two input sequences, and another for encoding the concatenation of the output from the first layer and the context vectors determined by the bi-directional attention mechanism. In our experiments, the dimensions of both layers are set to be the same, and the same type of recurrent unit is applied across both layers. The same settings are used for experiments on LSTM and our TPRU models. Specifically, for TPRU, the recurrent units in both layers have the same number of role vectors. Other learning details are as in the plain architecture. Tables 2 and 3 respectively present results on the Logical Entailment and the MNLI task, with BiDAF results in parentheses.

### C.3    LANGUAGE MODELLING:

On the language modelling task, the same two-layer architecture (Merity et al., 2018) is adopted in our experiments on LSTM and our proposed TPRU. For WikiText-103, Bytepair encoding (BPE) vocabulary with 100k merges (10k for WikiText-2) are applied to encode the corpus into indices (Sennrich et al., 2016). The output prediction layer is tied to the input word embedding layer as advised in prior work (Press & Wolf, 2017; Inan et al., 2016). Each model is trained for 64 epochs on the training set; the best one is selected based on perplexity on the validation set, and then evaluated on the test set. Table 4 shows the best performance of each model after the hyperparameter search.

## D    POLYSEMY

In the dev set of MNLI, the word bank occurs in the following sentences:

1. according to this plan , areas that were predominantly arab the gaza strip , the central part of the country , the northwest corner , and the west bank were to remain under arab control as palestine , while the southern negev desert and the northern coastal strip would form the new state of israel .

---

[5]https://gluebenchmark.com/leaderboard

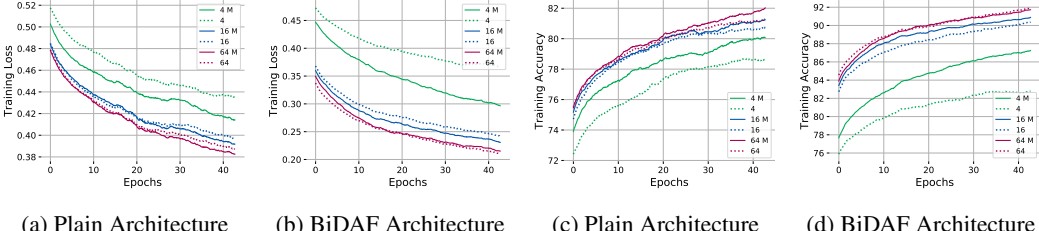

| (a) Plain Architecture | (b) BiDAF Architecture | (c) Plain Architecture | (d) BiDAF Architecture |

Figure 2: **Training Plots for comparing the single- and eight-slice dot-product operations of our TPRU, with both plain and BiDAF architectures.** (The plotted models are trained without dropout layers.) Overall, the eight-slice dot product, which enhances the fillers from one- to eight-dimensional vectors, leads to faster convergence rate and better performance on the training set when the number of role vectors is limited, and its effect diminishes when the number of role vectors reaches a large value. '4' and '4 M' respectively refer to our TPRU with four role vectors and single- or eight-slice dot-products. (Best viewed in colour.)

    2. well uh normally i like to to go out fishing in a boat and uh rather than like bank fishing and just like you try and catch anything that 's swimming because i 've had such problems of trying to catch any type of fish that uh i just really enjoy doing the boat type fishing .

    3. the other bank pays the fund interest based upon tiered account levels , more typical of a large commercial account .

The above sentences are three examples, there are other sentences in which the word bank occurs. As we show in the main paper, after learning, our TPRU categories the first two cases, in which one refers the West Bank of the Jordan river, and the other refers to the bank in the river context, as the same sense, as there is no clear learning signal that specifies the first one as a named entity, and categorises the third sentence into another sense.

## E    EFFECT OF INCREASING THE DIMENSIONALITY OF FILLERS/SYMBOLS

Both fillers and roles in TPRs (Smolensky, 1990) are tensors with arbitrary dimensions, which empowers the representation ability of TPRs. In our case, as we aim to produce a distributed vector representation of the given input sequence rather than a tensor representation, the fillers are reduced to filler numbers, in which case, the symbol represented by each filler is limited to one-dimensional vector.

In our proposed TPRU, the unbinding operation is instantiated by the dot-product (inner-product) operation between the binding complex and the unbinding vectors. One plausible way to upgrade filler numbers to filler vectors is to split the binding complex into multiple slices, and split each unbinding vector accordingly: then the dot-product is carried out on each pair of slices rather than the entire vector. In this case, the dimension of the resulting filler vectors is determined by the number of slices predefined prior to learning, and is limited by the dimension of the binding complex. The normalisation step is operated on the same dimension on multiple filler vectors independently and in parallel, which results in multiple normalised probability distributions. Each binding role vector is split into the same number of slices, and each distribution is applied to bind the same slices from all binding role vectors into a single slice. All slices are concatenated together to form a single vector which has the same dimensionality as the input binding complex. A similar idea was proposed as the multi-head attention mechanism in prior work and has been shown to be effective in terms of performance gain without additional computational cost (Vaswani et al., 2017). It was widely adopted in prior work on Transformer networks for various tasks.

In our implementation of the Logical Entailment task, the binding complex is split into eight slices, each with the same dimension. Figure 2 shows training logs with eight slices and also with only one slice for three different numbers of role vectors in our proposed TPRU. In general, shifting from filler numbers to filler vectors in our TPRU provides faster convergence and better performance on the training set. However, as the number of role vectors increases, the improvement gained decreases,

which leads to an interesting hypothesis that the effect of increasing number of role vectors might be correlated to that of increasing the dimensionality of filler vectors.

