# OpenReview forum: "A Simple Recurrent Unit with Reduced Tensor Product Representations"
_ICLR.cc/2020/Conference — Reject_

### Official Review · AnonReviewer3 · 2019-10-09
**Official Blind Review #3**

**Rating:** 6

**Review:**

This paper proposes a new recurrent unit with a simplified dynamics during training leading to more stable training algorithms and better performance. The paper is difficult to read because it assumes the reader is an expert on Tensor Product Representations. Many important terms are not clearly defined, which makes difficult to follow. For example, the terms “roles” and “filler” are not defined. I think a quick introduction to the field with a clarifying figure would be greatly appreciated by general readers. However, I think the contribution of the paper is important and presented experimental results, comparing the method against classical LSTM and GRU architectures, seem to be relevant for the field.







**Experience Assessment:**

I do not know much about this area.

**Review Assessment: Checking Correctness Of Derivations And Theory:**

I did not assess the derivations or theory.

**Review Assessment: Checking Correctness Of Experiments:**

I assessed the sensibility of the experiments.

**Review Assessment: Thoroughness In Paper Reading:**

I made a quick assessment of this paper.

---

### Official Review · AnonReviewer2 · 2019-10-17
**Official Blind Review #2**

**Rating:** 3

**Review:**

In this paper, a new unit based on the outer product called TPRU is proposed for recurrent neural networks. The performance of TPRU is validated with several NLP tasks such as POS tagging.

While my knowledge about RNNs is limited, I feel the paper has room for improvement and I vote for rejection this time. The main reasons are: 1. the paper is not well written, and 2. the way of analysis is not enough.

1. From the viewpoint of an RNN non-expert (i.e., me), the paper somehow fails to introduce the background. For example, tensor product representation (TPR) is introduced in the second paragraph of Introduction. While TPR is an elemental idea of this study, it is introduced with neither motivation (why and when TPR is useful, etc) nor appropriate references, which makes non-experts difficult to catch up with the main body of this study. So the paper is not self-contained enough.

2-1. The paper tries to explain why TPRU is better in terms of gradient vanishing/explosion in Section 4. However, the analysis is mainly performed in a qualitative way, and there is no quantitative analysis of it. For example, I expect something like the evaluation of the magnitude of the gradient, e.g., how much degree the gradient scale is reduced from normal gate to the TPRU gate. Or, at least there should be the numerical experiments for the comparison. Otherwise, it is hard to judge whether the gradient is actually stabilized.

2-2. The paper says one of the advantages of using TPRU is in its interpretability. However, the term "interpretability" is very vague and it is not properly defined in this paper. The paper should discuss what is the metric of interpretability here. More specifically, the paper claims TPRU's interpretability by Table 5. It looks, however, improper because there is no baseline and we cannot conclude that TPRU has better interpretability than others.

**Experience Assessment:**

I do not know much about this area.

**Review Assessment: Checking Correctness Of Derivations And Theory:**

N/A

**Review Assessment: Checking Correctness Of Experiments:**

I assessed the sensibility of the experiments.

**Review Assessment: Thoroughness In Paper Reading:**

N/A

---

### Official Review · AnonReviewer1 · 2019-10-24
**Official Blind Review #1**

**Rating:** 3

**Review:**

This paper proposes a novel model of recurrent unit for RNNs which is inspired from tensor product representation (TPR) introduced by Smolensky et al. in 1990. The authors claim that this allows one to better incorporate structural information into learning and easier interpretability for the learned representations. The proposed approach is motivated by a theoretical analysis showing that using TPR in this context acts as a sort of pre-conditioner and stabilizes learning. Experiments on entailment tasks (given two statement, decide whether the first implies the second) are provided to validate the approach.

I find the paper not easy to follow, with a non-negligible amount of typos in the notations and results. The advantage in terms of accuracy of the proposed approach seems marginal in the experiment, and the analysis of the interpretability of the learned representations could be improved: loosely speaking, particular examples of interpretability are given but sometimes without contexts or baselines to compare to (see the two last comments below). I know "interpretability" is a difficult property to assess but I think there may be more principled ways to showcase the approach.

I think this paper is not yet ready for publication: the proposed model is interesting and relevant but its validity could be better assessed and the paper needs some thorough proof-reading.


* Questions / Comments *

- page 1: the authors write U^TR = I, but I believe this is only possible if the TPR dimension d is bigger than the number of roles N. Is this always the case? This should be clarified.
- related to the previous point:  if U^TR = I then shouldn't U^Tb_{t-1} simply be f_{t-1} in Eq. 1?
- before Eq.1, f should be from R^d\times R^d' to R^d, not from R^d \times R^d
- In Eq. 1, b_{t-1} and x_t are not of the same dimension, so the cannot be multiplied by the same matrix U (this is why the matrices V_x and V_b are introduced later on).
- there seems to be a problem with Eq. (7): db_t/d_{b_{t-1}} appears on both sides of the equality...
- Modification 1: what is \tilde{vb_t}? I don't remember seeing this notation introduced before.
- Table 1: constants should not be included in big O notation! To compare constants, one should give the exact number of operations needed for inference.
- POS tagging: Aren't there many other reasons that could lead to this correlation (beside the informal argument that "TPR captures structured information")? Maybe the authors should compare with something else, for example the PMI between values of hidden neurons in a learned RNN and POS tags. Out of context, the numbers in Table 5 are not informative.
- Polysemy: Only a very specific cherry picked example is given here. A more principled or in depth analysis of this phenomenon is needed to make a stronger case.

* Typos *

- " The number of parameter matrices *is* the same as that of..."
- page 4 "stables" -> "stabilizes" (but rephrasing the sentence altogether would be better).
- page 5: BiDAF misses the capital letters ("bidaf").
- "dev set" -> "validation set" or "development set".
- page 8: "provides research*ers with* an intuitive..."?


**Experience Assessment:**

I have read many papers in this area.

**Review Assessment: Checking Correctness Of Derivations And Theory:**

I assessed the sensibility of the derivations and theory.

**Review Assessment: Checking Correctness Of Experiments:**

I assessed the sensibility of the experiments.

**Review Assessment: Thoroughness In Paper Reading:**

I read the paper at least twice and used my best judgement in assessing the paper.

---

### Author Response · Authors · 2019-11-15
**A new version is available. Thanks for the comments.**

Thanks sincerely for the comments and we have a new version uploaded.

---

### Decision · Program_Chairs · 2019-12-19

**Decision:**

Reject

**Comment:**

This paper has been reviewed by three reviewers and received scores such as 3/3/6. The reviewers took into account the rebuttal in their final verdict. The major criticism concerned the somewhat ad-hoc notion of interpretability, the analysis of vanishing/exploding gradients in  TPRU is experimental lacking theory. Finally,  all reviewers noted the paper is difficult to read and contains grammar issues etc. which does not help. On balance, we regret that this paper cannot be accepted to ICLR2020.